# Efficient Sleep Stage Identification Using Piecewise Linear EEG Signal Reduction: A Novel Algorithm for Sleep Disorder Diagnosis

**DOI:** 10.3390/s24165265

**Published:** 2024-08-14

**Authors:** Yash Paul, Rajesh Singh, Surbhi Sharma, Saurabh Singh, In-Ho Ra

**Affiliations:** 1Department of Information Technology, Central University of Kashmir, Ganderbal 191201, India; yashpaulcuk@gmail.com; 2Institute of Foreign Trade, New Delhi 110016, India; rajeshsinghin25@gmail.com; 3Department of Information Technology, National Institute of Technology, Srinagar 190006, India; surbhissjune@gmail.com; 4Department of AI and Big Data, Woosong University, Seoul 34606, Republic of Korea; singh.saurabh@wsu.ac.kr; 5School of Computer, Information and Communication Engineering, Kunsan National University, Gunsan 54150, Republic of Korea

**Keywords:** ADASYN, EEG, euclidean distance, halfwave, K-nearest neighbor, sleep states, SMOTE

## Abstract

Sleep is a vital physiological process for human health, and accurately detecting various sleep states is crucial for diagnosing sleep disorders. This study presents a novel algorithm for identifying sleep stages using EEG signals, which is more efficient and accurate than the state-of-the-art methods. The key innovation lies in employing a piecewise linear data reduction technique called the Halfwave method in the time domain. This method simplifies EEG signals into a piecewise linear form with reduced complexity while preserving sleep stage characteristics. Then, a features vector with six statistical features is built using parameters obtained from the reduced piecewise linear function. We used the MIT-BIH Polysomnographic Database to test our proposed method, which includes more than 80 h of long data from different biomedical signals with six main sleep classes. We used different classifiers and found that the K-Nearest Neighbor classifier performs better in our proposed method. According to experimental findings, the average sensitivity, specificity, and accuracy of the proposed algorithm on the Polysomnographic Database considering eight records is estimated as 94.82%, 96.65%, and 95.73%, respectively. Furthermore, the algorithm shows promise in its computational efficiency, making it suitable for real-time applications such as sleep monitoring devices. Its robust performance across various sleep classes suggests its potential for widespread clinical adoption, making significant advances in the knowledge, detection, and management of sleep problems.

## 1. Introduction

For humans, sleep is an essential day and nocturnal activity. While the muscles relax, blood pressure, heart rate, and metabolism all drop while you sleep. The body can replace damaged or expired cells and mend muscles and tissues under certain circumstances. Sleep is also necessary for the brain’s ability to store and organize memories. However, the pressures of contemporary city living have changed people’s lifestyles, which has added to the general lack of sleep. With so many people suffering from different sleep disorders, this problem has grown to be a major global health concern. For instance, 56% of Americans, 23% of Japanese, and even 26% of Italians claim to have sleeplessness. The most prevalent mental illness associated with insomnia is depression, as around 90% of depressed people have trouble sleeping [1]. As a result, researchers have extensively explored various signals or combinations thereof to understand mental disorders, although the most effective ones remain ambiguous [2]. However, Electroencephalography (EEG) signals have shown promising performance in many cases compared to other signals or their combinations. We focus solely on EEG signals in the proposed algorithm, acknowledging that future investigations may incorporate different biomedical signals to enrich our understanding. Sleep constitutes a significant aspect of human life, with individuals spending approximately one-third of their lifespan asleep [3]. Sleep is broadly categorized into three main stages: wakefulness, REM (Rapid Eye Movement) sleep, and NREM (Non-Rapid Eye Movement) sleep. Early studies in the 1950s revealed that sleep is a heterogeneous behavioral state characterized by continuous transitions between distinct states, whose nomenclature depends on biological measurements. For instance, a state might be called Slow-Wave Sleep (SWS) or synchronized sleep when analyzed using EEG characteristics, as quiet sleep when considering behavioral correlates, and as NREM when eye movements are considered [3]. The R&K classification system divides sleep into six categories: REM; sleep stages 1 through 4; and wakefulness, with stages 3 and 4 combined as SWS. Typically, NREM exhibits a higher magnitude than REM, with more regular breathing and heart rates. Detailed nightly recordings of biological signals and professional manual scoring are required, by recognized standards, to enable reliable classification of sleep cycles. However, manual scoring is time-consuming and costly, necessitating automatic sleep phase detection to enhance accessibility and reduce expenses. Despite the utility of EEG signals, their patterns in infants under one year of age are unstable [4]. The brain’s electrical activity is recorded by EEG, which shows various patterns in different frequency ranges. These patterns reflect various cognitive states as well as physiological functions. Delta waves, which have a frequency range of 0.5 to 4 Hz, are common in profound sleep and unconsciousness. Theta waves, ranging from 4 to 8 Hz, are frequently seen during sleep, dreaming, and some memory functions. When the human brain is idle during relaxation or meditation, the alpha waves usually predominate between 8 and 12 Hz. When thinking actively, solving problems, making decisions, and staying awake, beta waves, which have a frequency range of 12 to 30 Hz, are often present. They are further divided into three categories: high beta (20–30 Hz), mid beta (15–20 Hz), and low beta (12–15 Hz), each of which is linked to a different degree of cognitive activity. Waves between 30 and 100 Hz, known as gamma waves, are associated with complex mental functions, memory formation, perception, and awareness. These EEG frequency bands can be analyzed to learn important information about mental wellness, cognitive moods, and brain activity. Drowsiness while driving poses a significant risk, contributing to approximately 100,000 accidents annually in the USA, resulting in 1500 fatalities and 71,000 severe injuries [5]. Sleep deprivation and poor sleep quality are significant public health concerns, impacting cognitive performance, emotional regulation, and overall health. Polysomnography is a standard sleep state detection and monitoring method, particularly in diagnosing sleep-related disorders [6]. However, its manual scoring process is laborious, emphasizing the need for automated sleep state scoring systems. Our proposed method leverages a piecewise linear model to decompose EEG signals and extract features, subsequently employing classifiers such as K-nearest neighbors (KNN) for sleep state classification. Additionally, integrating machine learning algorithms with EEG signal analysis has the potential to enhance the accuracy and efficiency of sleep disorder diagnosis, leading to better patient outcomes and streamlined clinical practices.

## 2. Related Work

In the analysis of the literature, we looked at several linear and nonlinear sleep state detection methods. We discovered that to extract the components of the features vector from non-periodic signals like EEG, the recent research studies suggest and concentrate on parameters like correlation dimension, Lyapunov exponent, standard deviation, variance, approximate entropy, mode, mean, the energy of the signal, slopes, etc. Different studies suggest different signals or combinations to detect sleep states. Alshammari and Talal Sarheed [7] evaluated and optimized various machine learning algorithms for sleep disorder classification using the Sleep Health and Lifestyle Dataset, which comprises 400 rows and 13 features. Among k-nearest neighbors, support vector machines, decision trees, random forests, and artificial neural networks (ANNs), the ANN achieved the highest classification accuracy of 92.92%. Satapathy et al. [8] compare machine and deep learning algorithms for sleep disorder detection using EEG data, revealing that deep learning models, especially CNNs and RNNs, significantly outperform traditional methods. Deep learning models effectively capture complex patterns by analyzing spectral, temporal, and spatial EEG features, leading to superior accuracy, sensitivity, specificity, and precision in diagnosing sleep disorders. These findings highlight the potential of deep learning for early and accurate detection of conditions like insomnia, sleep apnea, and narcolepsy. The research proposed by Zhao et al. [9] examined various feature extraction and classification techniques for sleep staging and summarized the algorithms used in the literature along with the staging outcomes. In addition, a total of 22 features, such as Kurtosis, Skewness, Hjorth parameters, Standard Deviations, Wavelets Energy, Sample Entropy (SampEn), Fuzzy Entropy, Tsallis Entropy, Fractal Dimension (FD), and complexities, are listed in this paper based on the Time Domain, Time-Frequency, and Nonlinear analysis methods. They used the Physionet EDF database (Extended), and the algorithm is based on single-channel EEG data, where Wavelet Transform (WT) and Support Vector Machine (SVM) are utilized to achieve sleep staging. ANOVA was also used to assess the data on the distinctive features. Abbasi et al. [10] used a Convolutional Neural Network (CNN) to develop an automated and operationally effective technique for identifying neonatal Quiet Sleep (QS). A total 38 h of EEG recordings from 19 neonates at Shanghai, China’s Fudan Pediatric Hospital were used in the study. Twelve important time and frequency domain features from nine bipolar EEG channels were used to train and evaluate the CNN. Two convolutional layers with pooling and Rectified Linear Unit (ReLU) activation were part of the structure of CNN. A smoothness filtering was also applied to maintain the sleep state for three minutes. When contrasted to individual expert annotations, the suggested method performed admirably, achieving 94.07% accuracy, 89.70% sensitivity, 94.40% specificity, 79.82% F1-score, and a 0.74 kappa coefficient. Wang et al. [11] introduced a Sleep EEG Net model employing domain adaption transfer learning techniques to tackle issues with sleepiness detection. Transfer training is used by the model, which was pre-trained on the physioNet Sleep-EDF dataset, to facilitate cross-domain information transfer. An average identification accuracy of 91.5% in tiredness detection tests is found and the system exhibits consistency and strong generalization in simulated and real-world driving scenarios.

Three medical signals—namely, EEG, Electrooculogram (EOG), and Electromyography (EMG)—were employed by Jiang et al. [12]. This method used a leave-one-out cross-validation strategy while performing the training and testing. The method achieved an average accuracy of 81.2% and a Cohen’s Kappa coefficient of 0.722.

Nicola et al. [13] used electrical brain waves to identify sleep patterns automatically on only one channel. The time and frequency domain features are combined and categorized with an average accuracy of 90.81% and 83.2% for the first two and first four sleep stages, respectively. They obtained an accuracy of 86.7% overall. The variance characteristics or features from the various bands of the EEG signal and dispersion entropy were employed by Tripathy et al. [6,14]. The EEG information with the RR-time series parameters was supplied to a Deep Neural Network (DNN) to categorize each stage of sleep. With an overall accuracy of 85.51%, 94.03%, and 95.71%, they were able to classify sleep from awake, light sleep from deeper sleep, and REM from NREM sleep stages. The accuracy percentage overall was 91.71%. A single-channel technique utilizing wavelet transform to deconstruct the EEG signal was presented by Silveira et al. [15]. The wavelet coefficient’s characteristics, including their variance, skewness, and kurtosis, are categorized using a Random Forest classifier. For two to six classes, they achieved an accuracy of 90% overall. Budak et al. [16] introduced a novel technique for identifying driver fatigue. They use the Q-factor wavelet transform to break down the signal into smaller bands. Computation comprises the sub-band spectrogram images that were generated as statistical information like the current frequency and Standard Deviation (SD). For classification, Long Short-Term Memory (LSTM) is used. They achieved an overall accuracy of 94.31% for both the awake and sleepy (S1) phases. Hermite functions were used as base functions by Taran et al. [17]. Hermite coefficients are employed as features to classify states of alertness and drowsiness. Applying the Extreme Learning Method (ELM), their relative identification rate for awake and drowsy are 95.45% and 87.92%. The precision of the study as a whole was 92.28%. Twelve features were retrieved using three approaches in the subject-specific approach [18], and such features are heart rate variability (HRV), Detrended Fluctuation Analysis (DFA), and Windowed DFA (WDFA). They stated that their kappa coefficient was 0.43 and mean accuracy was 79.99. In the research of Barnes et al. [19], sleep apnea (SA) events were identified from only one-channel brain waves using an Explainable Convolutional Neural Network (ECNN). Three convolutional layers made up the CNN architecture, and the Hyperband method was used to adjust the hyperparameters for each layer. The network’s effectiveness was measured using ten-fold cross-validation, which produced a 69.9% precision and a 0.38 Matthews–Pearson Correlation coefficient (MCC). Critical-Band Masking (CBM) and lesioning analyses were used to understand the mechanisms of the network that was trained. Acharya et al. [20] collected two distinct datasets in their investigation: Overnight Polysomnography, comprising the EEG signals of 14 patients from University College Dublin, Ireland, and the Apnea Sleep Database of 25 subjects from St. Vincent’s University Hospital/University College Dublin. A thorough comparison of 29 nonlinear techniques and parameters for EEG-based sleep stage detection was provided in this research. It also demonstrated how HOS and RQA can be used to characterize different stages of sleep. Every nonlinear parameter yields clinically meaningful outcomes, meaning the measurements can distinguish between different sleep stages.

To analyze and automatically identify the different stages of sleep using EEG signals, Acharya et al. [21] employed HOS. HOS features are derived from the bispectrum and bicoherence plots of various sleep stages. A classification accuracy of 88.7% is achieved when significant features are supplied into a Gaussian mixture model classifier for classifying sleep stages. For other methods related to sleep states and the recommendations about different classifiers, readers are advised to read these references [22,23,24,25,26].

The literature review indicates that existing time and frequency analyses are insufficient for capturing detailed information in EEG signals due to their non-stationary and nonlinear nature. Nonlinear dynamics have proven more effective in differentiating sleep stages. Deep learning algorithms perform better than other methods. However, these advanced methods are often complex, slow, and still need accuracy improvements, making them unsuitable for real-time applications. To address these issues, a new approach is proposed that simplifies the input signals; these simplified signals are treated as piecewise linear functions in the time domain, offering a simple and fast solution compared to existing state-of-the-art methods.

## 3. Dataset Used

### Dataset

In this study, we utilized the well-established MIT-BIH Polysomnographic Database, curated and detailed by Ichimaru et al. [27], which originates from the Sleep Laboratory at Boston’s Beth Israel Hospital. This open-source database is readily accessible at https://www.physionet.org/physiobank/database/slpdb/ (accessed on 9 July 2024). The 39 overnight polysomnographic recordings from 20 participants, aged 23 to 63, make up the MIT-BIH Polysomnographic Database. Each recording lasts for seven to eight hours on average and includes respiratory signals like thoracic and abdominal movements, nasal airflow, and occasionally pulse oximetry, in addition to signals like EEG, EOG, EMG, and ECG.

These recordings were produced with a range of electrode arrangements and sampling frequencies. The signals were recorded at different sampling rates between 10 Hz and 256 Hz after being digitalized with a 12-bit resolution. Annotations for sleep phases (e.g., waking; NREM stages N1, N2, N3; and RE) and particular events (e.g., apneas, hypopneas, arousals, and leg movements) are included in the database. The MITBPD data distributions are 17.79%, 38.28%, 4.76%, 1.78%, 6.89%, and 30.5% for NREM1, NREM2, NREM3, NREM4, REM, and awake sleep stages, respectively. Different recordings of the database are shown in the following Figure 1.

These datasets consist of three channels: C3-O1, C3-A1, and O2-A1. The ECG and respiratory signals are labeled with sleep stages and apnea occurrences, and the electrocardiogram (ECG) signals are beat-by-beat. Notably, each signal segment is partitioned into 20 and 30 s epochs, with each epoch corresponding to a specific sleep stage. The signals were sampled at a rate of 250 Hz, and expert annotators labeled the 30-second duration of EEG and other signals. Our experimentation involved tests on eight records selected from the dataset, as outlined in Table 1, Table 2, Table 3, Table 4 and Table 5.

A major difficulty in EEG signal processing is choosing a specific channel from multiple channels. In the given database, every patient only had access to a single EEG channel. Therefore, we had no choice in channel selection. Unfortunately, the author of the database did not furnish any information regarding the process behind channel selection.

**Table 3 sensors-24-05265-t003:** Results of the proposed algorithm with the different number of classes.

Subject File	Classes	Sensitivity (%)	Specificity (%)	Accuracy (%)
slp01a	1,2,3,4,W,R	92.59	93.84	93.21
slp01b	1,2,W,R	96.86	98.78	97.82
alp2a	1,2,3,4,W,R	92.51	94.68	93.59
slp2b	1,2,W,R,M	94.34	95.77	95.05
slp03	1,2,3,W,R	96.42	97.73	97.075
slp04	1,2,3,W,R	94.59	97.73	96.16
slp14	1,2,3,4,W,R	94.42	96.75	95.58
slp16	1,2,3,4,W,R	96.84	97.92	97.38
**Avg**		**94.82**	**96.65**	**95.73**

**Table 4 sensors-24-05265-t004:** Comparing the outcomes with cutting-edge techniques considering all classes together tested on the same dataset.

Author and Year	Records Used	Classifier	Avg Accuracy (%)
Redmond et al. [28], 2003	17	QDA	76.75
Adnane et al. [18], 2012	17	SVM	79.99
Hayet et al. [29] 2012	09	ELM	83.59
Werteni et al. [30], 2015	17	SVM	56.81
Tripathy et al. [14], 2018	17	DNN	85.51, 94.0, 95.71
Taran et al. [17], 2018	16	ELM	92.28
Budak et al. [16], 2019	16	LSTM	94.31
An et al. [31],2019	06	W-SVM	85.29
Zhang et al. [32], 2020	18	CNN	87.6
Surantha et al. [33], 2021	18	SVM/ELM	76.77, 82.1
Rashidi et al. [1], 2023	18	DT	95.6, 92.72, 85.64
Wang et al. [34], 2023	18	GBDT	87.15, 82.02
**Proposed method**	8	KNN	**95.73%**

**Table 5 sensors-24-05265-t005:** Comparing the outcomes with cutting-edge techniques with different combinations of classes tested on the same database.

Author and Year	Number of Records	Features	Classes Used	Classifier Used	Average Accuracy (%)
Redmond et al. [28], 2003	17	HRV and EEG	W vs. REM vs. NREM	QDA	76,75
Adnane et al. [18], 2012	17	HRV, DFA, and WDFA	Sleep vs. wake	SVM	79.99
Hayet et al. [29], 2012	09	RR-time series and HRV	Sleep vs. wake	ELM	83.59
Warteni et al. [30], 2015	17	HRV	Sleep vs. wake REM	SVM	56.81
Tripathy et al. [14], 2018	17	Dispersion entropy and variance	wake vs. light, sleep vs. deep, sleep vs. REM	Neural network	91.71
Taran et al. [17], 2018	16	Hermite coefficients	alert (w) and drowsiness (s1)	ELM	92.28
Budak et al. [16], 2019	16	Spectrogram images and instanious frequencies	alert and drowsiness	LSTM	94.31
**Proposed method, 2 classes**	8	Halfwave	2 random classes	KNN	**96.6**
An et al. [31], 2019	06	Statistical features	NREM (s1–s4), REM, Wake	W-SVM	85.29
Zhang et al. [32], 2020	18	Hilbert Huang coefficients	REM, NREM, wake	CNN	87.6
**Proposed method, random 4 classes**	8	Halfwave features	random 4 classes	KNN	**95.96**
**Proposed method, all 6 classes**	8	Halfwave	Wake, Sleep (all), REM	KNN	**95.73**

## 4. Proposed Method

A literature review performed by us and various studies like Motamedi-Fakh et al. [35,36] and [14,15,16,17,18,19,20] helped us to identify shortcomings in the existing studies, inspiring researchers to explore adaptive methods. Our research is motivated by this, aiming to enhance the speed and accuracy of sleep state detection compared to current methods. We use Halfwave modeling as our suggested method for obtaining feature vector segments. The main concept of our method is utilizing piecewise linear function models for time-based data reduction. These models offer low complexity, facilitating efficient processing while preserving essential information for accurate sleep state detection. Figure 2 displays the suggested algorithm’s framework.

### 4.1. Time Domain: Halfwave Method

In the latter part of the 20th century, the Halfwave method gained popularity for detecting epileptic seizures using EEG signals. In this context, entities called “spikes” and “sharp waves” (SSWs) were used to denote seizure and non-seizure segments [37]. The key advantage of this approach lies in its ability to discern normal and abnormal patterns within lengthy signals efficiently. Various versions of Halfwave methods, employing different criteria to define and identify Halfwaves, have been proposed over time.

However, our **motivation** diverges from these aforementioned approaches. Rather than focusing on identifying individual spikes, we aim to simplify EEG signals by treating them as piecewise linear functions and eliminating extraneous or irrelevant details. To achieve this, we developed a novel Halfwave method characterized by its simplicity and speed. Unlike existing Halfwave methods, which are typically guided by three principles [37], our approach is based on a single guiding principle.

Initially, we compute the extremal points of the original signal and discard intermediate values. Subsequently, we construct a piecewise linear function using these extremal points. Since the extremal points alternate between minimum and maximum values, the resulting piecewise function resembles a waveform. We observe that during intervals exhibiting an increasing trend, the temporary decrease in individual maxima–minima intervals is negligible and does not significantly contribute to the sleep detection process. Consequently, these minor fluctuations are deemed unnecessary and eliminated from the Halfwave. A similar process is applied during decreasing signal tendencies.

This procedure constitutes the first level of Halfwave decomposition. It can be iteratively repeated, resulting in subsequent levels of Halfwave reduction. However, it becomes evident that after several iterations, no further changes occur, indicating that the next level of decomposition aligns with the previous one. This finalized decomposition is termed the “complete” Halfwave, while intermediate stages are referred to as semi-Halfwave decompositions.

As we move from lower to higher levels in the Halfwave reduction process, more signal details are lost. Thus, a critical aspect of the reduction problem lies in determining the most suitable level for the specific application, such as sleep detection in our case.

Following this informal overview, we provide the mathematical formalization of our Halfwave method.

#### 4.1.1. Mathematical Formalization of Proposed Halfwave Method

Let us consider a signal f:[a,b]→R that is continuous on the compact interval [a,b], with a finite number of extrema within this interval. We begin by categorizing these extrema into two sets:$$M_{1}:=\left\{x\in \left[a,b\right]\;:\;f\;\mathrm{has}\;a\;\mathrm{local}\;\mathrm{maximum}\;\mathrm{at}\;x\right\},$$
$$m_{1}:=\left\{x\in \left[a,b\right]\;:\;f\;\mathrm{has}\;a\;\mathrm{local}\;\mathrm{minimum}\;\mathrm{at}\;x\right\}.$$

The following represents the total set of all extremal points:(1)$$X_{1}:=M_{1}\cup m_{1}=\left\{x_{0},x_{1},x_{2},\ldots ,x_{n}\right\}$$
where the elements are arranged in increasing order:(2)$$x_{0}<x_{1}<x_{2}<\cdots <x_{n}\;\left(n\in N\right)$$
assuring a rotation between the minimum and maximum positions.

We remove some extremal points from $M_{k}$, $m_{k}$, and $X_{k}$ in each minimization step *k* of the proposed algorithm (k=1,2,3,…), keeping only the essential extremal points to build the new sets $M_{k+1}$, $m_{k+1}$, and $X_{k+1}=M_{k+1}\cup m_{k+1}$. Although we stop the procedure at a suitable iteration $k^{*}$, the algorithm converges, suggesting that there exists an iteration K∈N such that for every k≥K, $M_{k}=M_{K}$ and $m_{k}=M_{K}$.

A single algorithmic step consists of minimizing undesired extrema. With the elements of $X_{1}$ labeled in ascending order, we start with $M_{1}$, $m_{1}$, and $X_{1}=M_{1}\cup m_{1}$. Next, we clarify
(3)$$y_{i}=f\left(x_{i}\right)\;\left(i=0,\ldots ,n\right)$$
as the extremal values and
(4)$$\Delta_{i}:=y_{i+1}-y_{i}\;\left(i=0,\ldots ,n-1\right)$$
as the variations between two successive extreme values, a minimum and a maximum. The set of segment indexes where the differences are considered trivial is identified as follows:(5)$$D:=\{i\in N\;:\;1\leq i<n-1\;|\Delta_{i}|\leq |\Delta_{i+1}|\;\wedge \;|\Delta_{i}|\leq |\Delta_{i-1}|\}\cup \left\{n\right\}$$

These indexes represent segments where the differences are less than their neighbors. We then remove the endpoints of such segments from the set of extremal points. It is important to note that if i∈D, then i−1 and i+1 are not in *D*. Consequently, if $x_{i}$ and $x_{i+1}$ are removed, then $x_{i-1}$ and $x_{i+2}$ are retained, preserving the alternating extremal property. Thus, the set of extremal points in level 2 is defined as
(6)$$X_{2}=\left\{x_{i}\in X_{1}\;:\;i-1,\;i\notin D\right\}$$

The values in $X_{2}$ alternate between minimum and maximum points, while
$$M_{2}=X_{2}\cap M_{1},$$
$$m_{2}=X_{2}\cap m_{1}$$
represent the sets of new extremal values. This procedure can be repeated for $X_{2}$, $M_{2}$, and $m_{2}$ to obtain subsequent levels until the desired level is reached.

#### 4.1.2. Advantages of Proposed Halfwave Method

1.Efficiency in Processing: The method simplifies EEG and other signals by treating them as piecewise linear functions, which reduces data complexity while preserving essential information. This makes the Halfwave method efficient in processing lengthy signals.2.Low Complexity: By computing the extremal points of the original signal and constructing a piecewise linear function, the method reduces unnecessary data, thereby lowering the complexity of the analysis. This simplification helps in faster processing and real-time application.3.Accuracy: The Halfwave method has shown high accuracy in various applications, such as sleep state detection. It is particularly effective in distinguishing normal and abnormal patterns within signals, which is crucial for applications like epileptic seizure detection and other brain disorders.4.Adaptability: The method’s flexibility allows for iterative decomposition, which can be adjusted to find the most suitable level of detail for a specific application. This adaptability is essential for tasks like sleep detection, where different levels of signal detail may be required.

## 5. Features Extraction and Classification

### 5.1. Feature Extraction

The feature vector employed in our study is derived from EEG signals, each labeled by experts to categorize various sleep states within 30 s intervals. With a sampling rate of 250 Hz, each 30 s segment comprises 7500 sample points. We adopt the Halfwave method, limiting the reduction to level 2. This decision is informed by the observation that higher reduction levels lead to overly simplified signals, with some segments containing only one extremal point. Consequently, such reduced signals lack sufficient information for effective feature extraction, particularly evident in the early stages due to the nature of sleep state signals—characterized by slow activities and less pronounced peaks. One advantage of the Halfwave method lies in its adaptability to specific problems, allowing customization as needed.

Our chosen time domain features include the **total number of extremal points, slopes of linear segments, maximum slope, mean of extremal points, absolute minimum, and maximum within the window**. The mathematical formulation of the different features is given below:

1. **Total Number of Extremal Points (E)**:E=totalnumberoflocalmaximaandminimainthedata

2. **Slopes of Linear Segments ($S_{i}$)**: $S_{i}=\frac{y_{i+1}-y_{i}}{x_{i+1}-x_{i}}\;\mathrm{for}\;\mathrm{each}\;\mathrm{segment}\;i=1,2,\ldots ,n-1,$ where $\left(x_{i},y_{i}\right)$ and $\left(x_{i+1},y_{i+1}\right)$ are the coordinates of consecutive points.

3. **Maximum Slope ($S_{\max}$)**: $S_{\max}=\max\left(S_{1},S_{2},\ldots ,S_{n-1}\right)$

4. **Mean of Extremal Points ($\bar{E}$)**: $\bar{E}=\frac{1}{E}\sum_{i=1}^{E}y_{e_{i}}$ where $y_{e_{i}}$ are the y-values of the extremal points.

5. **Absolute Minimum ($y_{\min}$)**: $y_{\min}=\min\left(y_{1},y_{2},\ldots ,y_{n}\right)$

6. **Absolute Maximum ($y_{\max}$)**: $y_{\max}=\max\left(y_{1},y_{2},\ldots ,y_{n}\right)$

These features have demonstrated efficacy in sleep state detection, supported by various studies and surveys. Our selection process involved extracting numerous statistical features in the time domain, followed by analysis via histograms. This analysis revealed that the identified **six-time domain features** offer the most discriminative power for our task.

### 5.2. Classification

After building the feature vector from Halfwave, we classified the feature vector by using the K-Nearest Neighbor algorithm (KNN), which proved to be more effective in our study—as shown in Table 1 and Table 2)—than other algorithms like the support vector machine (SVM), decision tree, random forest, and artificial neural network (ANN). The literature review revealed that KNN, ANN, CNN, and SVM are commonly used classifiers for this task. KNN is widely used because it is non-parametric in nature, instance-based, straightforward, resilient, adaptable, quick, and a supervised classifier. The results from different classifiers (for example, here, we have shown the results only for SVM and KNN but tested other classifiers too) are shown in Table 1 and Table 2. To find the best classifier initially, we have chosen 6 h long EEG signal data from the five different records (description shown in Table 1 and Table 2) as sample data. Different classifiers are trained on these data with 60–40 training and testing strategy. The KNN classifier performs better than well-known classifiers. Later, we applied our proposed method to the available EEG signal data for eight records. In this research work, the value of K in KNN is considered as 2 and the distance metric used here was an advanced version of Euclidean distance [38], which is faster than traditional Euclidean distance, as explained in the coming sections. The mathematical background of the KNN along with parameter tuning is explained below.

#### 5.2.1. K-Nearest Neighbors Classification

#### Data Representation

Let $D={\left\{\left(x_{i},y_{i}\right)\right\}}_{i=1}^{N}$ be the training dataset, where

$x_{i}\in R^{d}$ is the *i*-th feature vector with *d* dimensions;$y_{i}\in \left\{1,2,\ldots ,C\right\}$ is the class label corresponding to $x_{i}$;*N* is the total number of training samples;*C* is the number of distinct classes.

#### Distance Metric

The most commonly used distance metric in KNN is the Euclidean distance. For two feature vectors $x_{i}$ and $x_{j}$, the Euclidean distance $d(x_{i},x_{j})$ is given by
$$d\left(x_{i},x_{j}\right)=\sqrt{\sum_{k=1}^{d}{\left(x_{ik}-x_{jk}\right)}^{2}}$$
where $x_{ik}$ and $x_{jk}$ are the *k*-th components of the feature vectors $x_{i}$ and $x_{j}$, respectively.

#### Algorithm Steps

Step 1: Compute Distances

For a given test sample $x_{\mathrm{test}}$, compute the distance from $x_{\mathrm{test}}$ to each training sample $x_{i}$ in the dataset D:$$d\left(x_{\mathrm{test}},x_{i}\right)=\sqrt{\sum_{k=1}^{d}{\left(x_{\mathrm{test},k}-x_{ik}\right)}^{2}}$$

Step 2: Identify Nearest Neighbors

Sort the distances in ascending order and select the *K* training samples with the smallest distances. Let these *K* nearest neighbors be denoted by $N_{K}\left(x_{\mathrm{test}}\right)$.

Step 3: Vote for the Class Labels

Each of the *K* nearest neighbors vote for their respective class labels. Let $y_{j}$ be the class label of the *j*-th nearest neighbor. Count the votes for each class. The predicted class label ${\hat{y}}_{\mathrm{test}}$ for the test sample $x_{\mathrm{test}}$ is the class with the majority vote:$${\hat{y}}_{\mathrm{test}}=\arg\max_{c\in \{1,2,\ldots ,C\}}\sum_{i\in N_{K}\left(x_{\mathrm{test}}\right)}I\left(y_{i}=c\right)$$
where $I(y_{i}=c)$ is the indicator function, which is 1 if $y_{i}=c$ and 0 otherwise.

#### Formal Definition

Given a test sample $x_{\mathrm{test}}$, the KNN classification rule can be formally defined as
$${\hat{y}}_{\mathrm{test}}=\arg\max_{c\in \{1,2,\ldots ,C\}}\sum_{i\in N_{K}\left(x_{\mathrm{test}}\right)}I\left(y_{i}=c\right)$$ Here, $N_{K}\left(x_{\mathrm{test}}\right)$ represents the indices of the *K* nearest neighbors of $x_{\mathrm{test}}$ in the training set.

#### Example Illustration

For instance, if K=3, and the three nearest neighbors of a test sample $x_{\mathrm{test}}$ have class labels {2,1,2}, the predicted class label would be
$${\hat{y}}_{\mathrm{test}}=\arg\max_{c\in \{1,2\}}\left(\sum_{i\in \{1,2,3\}}I\left(y_{i}=c\right)\right)$$ In this case, class 2 has the majority vote, so
$${\hat{y}}_{\mathrm{test}}=2$$

In our proposed method, we use an Extended version of Euclidean distance computations, as mentioned above. In the coming subsections, we have shown how an extended version is faster than simple Euclidean distance computation.


**Traditional computation of Euclidean Distance Matrix**


Suppose we have a collection of vectors $\left\{x_{i}\in R^{d}:i\in \left\{1,\ldots ,n\right\}\right\}$ and we want to compute the n×n matrix, *D*, of all pairwise distances between them. We first consider the case where each element in the matrix represents the squared Euclidean distance, a calculation that frequently arises in machine learning. The distance matrix is defined as follows:(7)$$D_{ij}={∥x_{i}-x_{j}∥}_{2}^{2}$$
or equivalently,
(8)$$D_{ij}={\left(x_{i}-x_{j}\right)}^{T}\left(x_{i}-x_{j}\right)=∥x_{i}{∥}_{2}^{2}-2x_{i}^{T}x_{j}+{∥x_{j}∥}_{2}^{2}$$

There is a popular “trick” for computing Euclidean Distance Matrices (although it is perhaps more of an observation than a trick). The observation is that it is generally preferable to compute the second expression rather than the first.

Writing $X\in R^{d\times n}$ for the matrix formed by stacking the collection of vectors as columns, we can compute Equation (Equation 7) by creating two views of the matrix with shapes of d×n×1 and d×1×n, respectively.


**Basic steps for naive Computation of Euclidean Distance Matrix**



**Input:**

X∈Rd×n

1.A←reshape(X,(d,n,1)).2.B←reshape(X,(d,1,n)).3.$C\leftarrow A-B\in R^{d\times n\times n}$.4.$D\leftarrow 1_{d}^{T}C$.



**Naive Computation of Euclidean Distance Matrix along with storage and MACs overhead**



**Basic steps**

**Storage**

**MACs**

**Input:**

X∈Rd×n



d×n

-

A←reshape(X,(d,n,1))

--

B←reshape(X,(d,1,n))

--

$C\leftarrow A-B\in R^{d\times n\times n}$



d×n×n



d×n×n



$D\leftarrow 1_{d}^{T}C$



n×n×n



d×n×n


**Total**


$n^{2}\left(d+1\right)+nd$



$2dn^{2}$




**Notes:**


The reshape operation only changes the view of the data without altering the underlying memory layout. The total number of Multiply–Accumulate Operations (MACs) is 2dn.


**Basic steps of expanded computation of Euclidean Distance Matrix used in our model**



**Input:**

X∈Rd×n



1.$G\leftarrow X^{T}X\in R^{n\times n}$.2.$D\leftarrow \mathrm{diag}\left[G\right]+\mathrm{diag}{\left[G\right]}^{T}-2G$.


**Expended Computation of Euclidean Distance Matrix along with storage and MACs overhead**



**Basic steps**

**Storage**

**MACs**

**Input:**

$X\in R^{n\times n}$



d×n

-

$G\leftarrow X^{T}X\in R^{n\times n}$



n×n



d×n×n



$D\leftarrow \mathrm{diag}\left[G\right]+\mathrm{diag}{\left[G\right]}^{T}-2G$



n×n



2×n×n


**Total**


$2n^{2}+dn$



$n^{2}\left(d+2\right)$




**Observations:**


The matrix *G* is often referred to as the Gram matrix. The diag[G] operation selects the diagonal elements from *G* and stores them into an n×1 vector. Broadcasting is used in the final line to sum the vectors into a square matrix. Algorithm 1 requires approximately twice as many MACs as Algorithm 2 for most values of *n* and *d*. Storage costs are higher in Algorithm 1 due to the $n^{2}d$ term.

The reshape operation only changes the view of the data without altering the underlying memory layout. The total number of Multiply–Accumulate Operations (MACs) is 2dn. For more details about the advanced version of the Euclidean Algorithm, readers are requested to kindly refer to the work of Samuel Albanie [38].
**Algorithm 1** Naive Computation of Euclidean Distance Matrix1:**Input:**$X\in R^{d\times n}$2:D←ZeroMatrix(n,n)       ▹ Initialize the distance matrix3:**for**i=1**to***n***do**4:    **for** j=1**to***n* **do**5:        sum←06:        **for** k=1**to***d* **do**7:           diff←X[k,i]−X[k,j]  ▹ Compute difference between vectors8:           $\mathrm{sum}\leftarrow \mathrm{sum}+{\mathrm{diff}}^{2}$    ▹ Accumulate squared differences9:        **end for**10:        D[i,j]←sum          ▹ Store the squared distance11:    **end for**12:**end for**13:**Output:***D*


**Algorithm 2** Expanded Computation of Euclidean Distance Matrix
1:
**Input:**

$X\in R^{d\times n}$

2:G←ZeroMatrix(n,n)            ▹ Initialize the Gram matrix3:
**for**

i=1

**to**
*n*
**do**
4:    **for** j=1**to***n* **do**5:        $G\left[i,j\right]\leftarrow \sum_{k=1}^{d}X\left[k,i\right]\times X\left[k,j\right]$      ▹ Compute the Gram matrix6:    **end for**7:
**end for**
8:D←ZeroMatrix(n,n)           ▹ Initialize the distance matrix9:
**for**

i=1

**to**
*n*
**do**
10:    **for** j=1**to***n* **do**11:        D[i,j]←G[i,i]+G[j,j]−2×G[i,j]   ▹ Compute the distance matrix12:    **end for**13:
**end for**
14:
**Output:**
*D*




### 5.3. Class Balancing

The provided database contains an imbalanced collection of sleep state classes, and an unbalanced dataset will cause partiality in the classifier model. As a consequence, the issue of class imbalance must be resolved before using specific classifiers to avoid skewed outcomes. Two widely used approaches [39]—namely, under- and over-sampling—were used to address the class imbalance problem. Since there is no loss of data, over-sampling was employed in the majority of pattern categorization techniques. The suggested approach uses a sophisticated variant of the popular over-sampling method SMOTE (Synthetic Minority Oversampling Technique) [40] to address the issue of class inequalities in sleep states. This technique is known as the ADASYN (Adaptive Synthetic Sampling) [41], and it does not overfit or minimize the extracted features’ Receiver Operating Characteristic (ROC) curve [42]. Although this method is intended for two-class problems, we have also applied it to multi-class problems in which each class is evenly distributed concerning the class with the greatest number of training tuples. SMOTE: For each sample in the class, it first determines the n-nearest neighbors in the minority class. After that, it creates a line connecting the neighbors and places random points along it. ADASYN: This is a better Smote variant. It adds an arbitrary little value to the randomly generated points (random points in SMOTE) to make them more realistic. Put differently, rather than all the samples having a linear correlation with the parent, some of those have slightly greater variance, which is why they were scattered. Different classes (S1 (stage1), S2 (stage2), S3 (stage3), S4 (stage4), W (awake), and R (REM), based on the intensity or depth of sleep) we used to test our proposed algorithm are mentioned in the second column of Table 3.

## 6. Experimental Results

The proposed algorithm is tested on the physioNet POLYSONOGRAPGY dataset. The metrics listed below are used to assess the suggested algorithm’s efficiency:$$\begin{array}{c} \mathrm{Sensitivity}=\frac{\mathrm{TP}}{\mathrm{TP}+\mathrm{FN}}\times 100\;\;\\ \mathrm{Specificity}=\frac{\mathrm{TN}}{\mathrm{TN}+\mathrm{FP}}\times 100\;\;\\ \mathrm{Average}\;\mathrm{Accuracy}=\frac{\mathrm{TP}+\mathrm{TN}}{\begin{array}{c} \mathrm{TN}+\mathrm{FP}+\mathrm{TP}+\mathrm{FN} \end{array}}\times 100\; \end{array}$$

In a confusion matrix, the entities TP, TN, FP, and FN represent different aspects of the performance of a model:

When the classifier correctly identifies instances of the positive class, these are termed true positives (TP). When the classifier correctly identifies instances belonging to the negative class, they are referred to as true negatives (TN). False Positives (FP) occur when the classifier inaccurately predicts instances belonging to the positive class. False Negatives (FN) represent instances where the model incorrectly predicts the negative class.

For multi-class problems, these terms can be interpreted as follows: TP: The sum of the diagonal values of the confusion matrix represents the total number of correct predictions for all classes. TN: the total of every single column and row, except those corresponding to the class being considered, representing the total number of correct rejections for all classes. FP: The total number of incorrect positive predictions for all classes is given by the sum of the numbers in the relevant columns, avoiding TPs. FN: The total number of incorrect negative predictions for all classes, calculated as the total of the numbers in the corresponding rows minus true positives (TPs).

**Sensitivity:** Also called true positive rate, it calculates the percentage of real positive cases that the algorithm accurately detected. The average sensitivity of the mentioned approach is reported as 94.82% and is shown in Table 3.

**Specificity:** Specificity, also considered as the true negative rate, gauges how many legitimate negative cases the algorithm properly discovered. In Table 3, the average specificity of the proposed algorithm is reported as 96.65%.

**Accuracy:** When evaluating a method’s forecasts or predictions, accuracy needs to consider two entities—namely, true positives and true negatives—to find or know the overall accuracy. The computation involves dividing the overall number of correct projections by the entire number of predictions made by the algorithm. Table 3 reports the suggested algorithm’s average accuracy as 95.73%. Table 4’s 13th row illustrates the accuracy of the proposed method.

These performance metrics provide insights into the capabilities of the above-cited approach (proposed one) in detecting sleep states accurately. The high values of sensitivity, specificity, and accuracy indicate that the algorithm performs well in classifying sleep stages, as demonstrated by the results presented in Table 1, Table 2, Table 3, Table 4 and Table 5. The suggested algorithm’s performance is contrasted with that of other cutting-edge algorithms shown in Table 5, and we have seen that our novel approach is better than other latest techniques in terms of average accuracy.

## 7. Conclusions and Future Work

The novel approach presented in this study uses time-domain information extracted from the piecewise linear version of the EEG signal called Halfwave. The key concept of the piecewise linear model is to transform the signals so that they become simpler and smoother while retaining essential characteristics related to sleep states. Time domain features are calculated and analyzed before constructing the final feature vector, which is then used for classification. The proposed algorithm is evaluated on an extensive dataset comprising more than 70 h of data from the MIT-BIH Polysomnography database. Results demonstrate promising performance, with average sensitivity, specificity, and accuracy reported with all classes together considered at 94.82%, 96.65%, and 95.73%, respectively.

Moving forward, future research will focus on analyzing results obtained by combining features from multiple types of biomedical signals. Future research will include studies on clinical and experimental biomedical signals across a range of physiological and pathological conditions to evaluate the strengths and weaknesses of the proposed algorithm fully. By examining these diverse conditions, we can better understand the algorithm’s robustness and potential limitations, ensuring it performs well in various real-world scenarios. This approach aims to enhance the understanding of sleep phases and enhance the precision of sleep state detection algorithms.

## Figures and Tables

**Figure 1 sensors-24-05265-f001:**
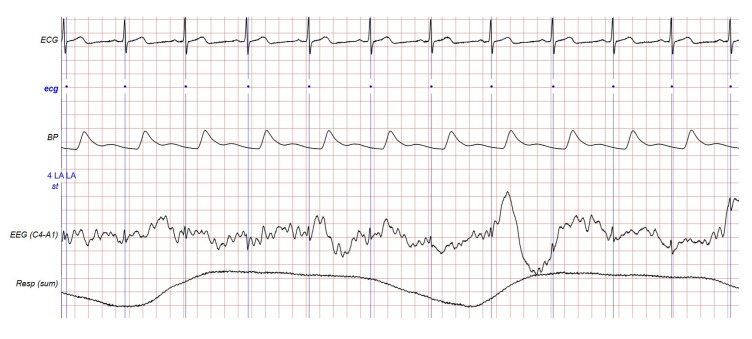
Different signals captured in the MIT-BIH Polysomnographic Database.

**Figure 2 sensors-24-05265-f002:**
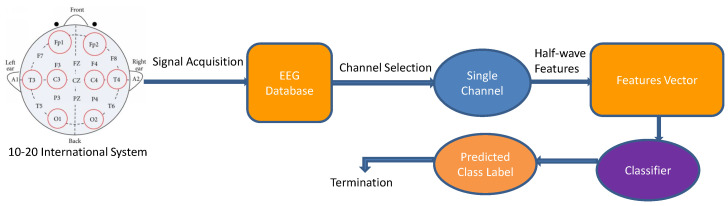
The Structure of the suggested approach.

**Table 1 sensors-24-05265-t001:** Classification on sample data using SVM and EEG signals.

Signal, Record and Duration	Classifier	Features	Train and Test	Sensitivity (%)	Specificity (%)	Accuracy (%)
EEG, slp01a, 90 mnts	SVM	Halfwave	60–40	92.36	91.42	91.89
EEG, slp01b, 60 mnts	SVM	Halfwave	60–40	60.87	89.79	75.33
EEG, slp2a, 90 mnts	SVM	Halfwave	60–40	39.60	80.60	63.1
EEG, slp2b, 60 mnts	SVM	Halfwave	60–40	80.36	90.30	79.21
EEG, slp03, 90 mnts	SVM	Halfwave	60–40	80.36	90.30	79.25

**Table 2 sensors-24-05265-t002:** Classification on sample data using KNN and EEG signals.

Signal, Record and Duration	Classifier	Features	Train and Test	Sensitivity (%)	Specificity (%)	Accuracy (%)
EEG, slp01a, 90 mnts	KNN	Halfwave	60–40	97.12	96.63	97.61
EEG, slp01b, 60 mnts	KNN	Halfwave	60–40	90.64	95.40	93.02
EEG, slp2a, 90 mnts	KNN	Halfwave	60–40	94.86	97.72	96.29
**EEG, slp2b, 60 mnts**	KNN	Halfwave	60–40	96.33	97.75	97.04
EEG, slp03, 90 mnts	KNN	Halfwave	60–40	95.73	96.48	96.10

## Data Availability

Data are contained within the article.

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
