# Peer review of "Efficient Sleep Stage Identification Using Piecewise Linear EEG Signal Reduction: A Novel Algorithm for Sleep Disorder Diagnosis"

_sensors, 2024, doi:10.3390/s24165265_

Round 1
Reviewer 1 Report
Comments and Suggestions for Authors
1. Many typos exist in the manuscript, e.g. "specificity, and Accuracy of the proposed algorithm on the Polysomnographic Database
considering 8 records is estimated as 94.82%, 96.65%, and 95.73%, Respectively". Why "A" is capital for the "accuracy"? Why "R" is capital for the "respectively"?
2. Wrong term. "to modern classifiers like ANN and SVM". The classifiers are not modern. These classifiers are common classifiers.
3. Comparison of the results on different datasets has no meaning. Table 6 has no meaning.
4. Typos prevail. "Saadullah et al. [ 6 ], The study suggest". The author Saadullah is absent. No one is writing like "Saadullah et al. [ 6 ], The study suggest". No need for "The study".
5. " [ 7],this work ". No one is writing this way.
6. "Two convolutional layers with pooling and rectified linear unit (ReLU) activation were part of the structure of CNN.". To which study is related this statement? The statement starts a new paragraph.
7. "were employed by Dihong et al. [ 8]. ". Such author is absent.
8. The review of related work must end on the summary showing deficiencies of related works and prospect for the new research.
9. The writing of review of related work is total collapse. The section is full of wrong author names, typos, incorrectly written statements. No summary at the end of the section to show shortcomings of related works and prospect for the new research. It seems like the authors are writing the review of related work to be published in highly respected journal for the first time in their life.
10. "3.2. Channel Selection"". Why to name "Channel selection", if there was no channel selection?
11. "Shayan et al. [17 ] identified shortcomings ":
a. Wrong naming of authors continues in the section of Proposed method.
b. It was your duty to identify the shortcomings of current related works in the section of Related Works, but you have not done this, since the study [17] is of the year 2014 and the current situation is different.
12. The writing style demonstrated in the manuscript does not suit to the publication to be published in the highly respected journal.
Extensive editing of English language is required
Reviewer 2 Report
Comments and Suggestions for Authors
Reviewer 1
This paper introduces a new algorithm for sleep disorder diagnosis and contributes to the growing knowledge of applying Piece-wise Linear analysis of EEG signals. The authors have done an excellent job proving the worth of new algorithms for classifying sleep stages. The work deserves to be published. Before that, however, I have some suggestions and questions that I think could further improve the quality of the work.
Comment 1
Perhaps the authors could consider adding and commenting on some literature data on the nonlinear analysis approaches to automated sleep stage detection in the context of their study. For instance, Acharya, U. R., Bhat, S., Faust, O., Adeli, H., Chua, E. C. P., Lim, W. J. E., & Koh, J. E. W. (2016). Nonlinear dynamics measures for automated EEG-based sleep stage detection. European Neurology, 74(5-6), 268-287; Zhao, D., Wang, Y., Wang, Q., & Wang, X. (2019). Comparative analysis of different characteristics of automatic sleep stages. Computer methods and programs in biomedicine, 175, 53-72. In other words, adding references and discussing them with the author's results might illuminate the promises and perils of their algorithm.
Comment 2
In the related work section, the authors mention a few critical computational approaches to detecting wake/sleep stages, including the Pearson correlation coefficient and their efficiency in detecting various sleep-related pathologies. Indeed, a simple statistical approach to EEG signals, such as the Pearson coefficient, could reveal pathological coupling between the EEG oscillations, such as sigma (in rats) or alpha in humans, and theta, which are the hallmarks of NREM and REM stages. Coupling changes in cortical and pontine sigma and theta frequency oscillations following monoaminergic lesions in the rat. Perhaps authors might consider discussing this issue in this section.
Comment 3
The authors conclude with the following sentence: "Moving forward, future research will focus on analyzing results obtained by combining features from multiple types of biomedical signals." I agree fully. However, the authors might add that future studies on clinical and experimental biomedical signals across physiological and pathological conditions will show all the strengths and weaknesses of the proposed algorithm.
Author Response
Please take a look at the attachment. The introduction section of the study is also improved

Reviewer 3 Report
Comments and Suggestions for Authors
1. Please provide a mathematical representation for KNN classification. Too few mathematical expressions throughout the paper.
2. Please explain why the Halfwave method is good and give the reasons.
3. How does it perform on self-collected dataset?
Comments on the Quality of English LanguageIt should be improved.
Round 2
Reviewer 1 Report
Comments and Suggestions for Authors
1. No need for the word "Summary" in the last paragraph of related work.
2. You are advised to add author names before reference numbers in table 4 and table 5.
Comments on the Quality of English LanguageModerate editing of English language is required
Author Response
Comment 1:No need for the word "Summary" in the last paragraph of related work
RESPONSE:
Respected sir/mam, thank you for giving us constructive feedback by reading our manuscript thoroughly. We have removed the word summary from the section.
Comment 2: You are advised to add author names before reference numbers in table 4 and table 5.
RESPONSE:
Thank you for your valuable feedback. We appreciate your suggestion to add author names before reference numbers in Table 4 and Table 5. We are pleased to inform you that we have already made these adjustments in the revised manuscript.
Note: We are working hard to improve the English language in the manuscript, and we are sure very soon, we will complete that part too.
Thank you once again for your insightful comments.
Best regards,
Dr Yash and the team
Reviewer 3 Report
Comments and Suggestions for Authors
I have no comments
Comments on the Quality of English LanguageIt can be improved.
Author Response
Dear Reviewer,
Thank you for your thorough review and valuable feedback. We are carefully revising the manuscript to improve the English language and overall readability. We also seek assistance from a native English speaker to ensure clarity and fluency throughout the text.
We appreciate your insights and hope that the revisions meet your expectations.
Best regards,
[Dr Yash and the Team